# Implementing Artificial Intelligence in Higher Education: Pros and Cons from the Perspectives of Academics

**Alina Iorga Pisica** [1], **Tudor Edu** [2], **Rodica Milena Zaharia** [1,*] **and Razvan Zaharia** [3]

[1] Department of International Business and Economics, The Bucharest University of Economic Studies, 050711 Bucharest, Romania
[2] Department of Marketing Management, Romanian American University, 012101 Bucharest, Romania
[3] Department of Marketing, The Bucharest University of Economic Studies, 050711 Bucharest, Romania
[*] Correspondence: milena.zaharia@rei.ase.ro

**Abstract:** This article investigates the perspectives of Romanian academics on implementing Artificial Intelligence (AI) in Higher Education (HE). The article analyzes the pros and cons of AI in HE, based on the views of eighteen academics from five Romanian universities. There is a large and heated debate about the proliferation of AI in many domains, with strong supporters and determined deniers. Studies that research the implications of AI enrich the evidence-based literature on the advantages, disadvantages, threats, or opportunities that AI creates for us, for businesses, or for societies. Though many aspects are still less well known, attitudes toward AI are still under construction. HE is a domain where the implications of AI create passionate discussions. HE is, eventually, the sector that shapes the masterminds of societies' leaders. There is a quest to find the perspectives of those who will apply AI, who will work with or for AI, and those who are opposed to or in favor of implementing AI in HE. The conclusions revealed by this study are in line with similar studies that exist in the literature. The positive aspects of AI implementation in HE are related, in the view of academics, to gains in the learning–teaching process, improvements in students skills and competences, better inclusion, and greater efficiency in administrative costs. Similarly, the negative aspects revealed by the research are linked to psychosocial effects, data security, ethical aspects, and unemployment threats. However, there are some aspects (mostly negative) related to implementing AI in HE that are less exposed by the interviewed academics, which are mostly related to the costs and efforts of implementing AI in HE. The possible explanation of this situation is related to the lack of strategic vision on what, in fact, the implementation of AI in HE means, what this process involves, and the fact that digitalization in Romanian universities (as well as in the Romanian economy) is in its infancy. The contribution of the results of this research is mainly empirical and practical. These opinions should be used as resources for managers of HE institutions to develop better policies concerning the implementation of AI in HE and for strategic vision toward AI, with the ultimate purpose of achieving progress and prosperity for the entire society.

**Keywords:** artificial intelligence; Higher Education; human and technological development; academics; pros; cons; Romania

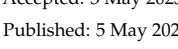

## 1. Introduction

New technologies pose huge challenges toward society. Artificial Intelligence detaches itself as a spearhead of these challenges, in terms of benefits and opportunities, or as inconveniences and threats toward all levels: from individuals, to businesses, from countries to regions and global community. The debates become hotter and hotter, as technology advances faster and faster, and the understanding, acceptance and/or adaptation to technological progress are not linear and lag behind technological developments. There is not a single area where the implementation of AI is not passionately debated. Higher Education, given its complex role in educating and building sustainable development, is at the center of the debate impact of and about the response toward AI.

This paper investigates the perspectives of Romanian academics on implementing AI in HE by analyzing the pros and cons, which emerged from in-depth interviews with eighteen academics from five Romanian universities. Based on qualitative research, the article is exploratory in nature. The findings are important for developing university strategies, as their competitive advantages depend on the implementation of new technologies in the educational process [1,2]. The results of this research will add to the growing literature on the topic and present an interesting case of a country (Romania) that has excellent internet endowments, but lower levels of digitalization of the economy [3,4]. This article starts with a brief literature review on the implementation of AI in HE. It continues with the methodological aspects and the limits of the research. The findings and results are presented after that. The paper ends with conclusions and recommendations for HE managers.

## 2. Theoretical Background

Artificial Intelligence (AI) derives from computer science [5,6] as a distinctive field of study, "the only field to attempt to build machines that will function autonomously in complex, changing environments" ([5], p. 18). Since the term "Artificial Intelligence" was first introduced in a workshop at Dartmouth College in 1956 ([5], p. 17), scientists and researchers have shown interest in the opportunities and threats that technology can give rise to in all aspects of our lives: from education to employment, and from social interactions to the very existence of us as a species. There is a huge amount of literature dedicated to Artificial Intelligence: the term "Artificial Intelligence" exposes over 580 million entries on Google in less than 0.30 s and over 4.5 million entries in 0.15 s on Google Scholar.

There has been a lot of debate about implementing Artificial Intelligence in education for more than 30 years, with huge investments in research and an AI education market estimated to reach over USD 25.7 billion in 2030 ([7], p. 1). The research areas cover at least two fundamental directions: "the science and engineering of making intelligent machines and intelligent computer programs" (the very definition of AI in the vision of one of the most prominent scholars in AI, John McCarthy ([8], p. 2)), and the implementation of AI, with a very wide range of aspects, such as "what" to implement, "how" to implement, the consequences of implementation, controlling implementation, or, as scholars have pointed out: "Learning with AI", "Using AI to learn about learning", "Learning about AI", and "Preparing for AI" ([9], pp. 19–21).

### 2.1. Benefits of Implementing Artificial Intelligence in Higher Education

Studies devoted to the benefits AI brings to education in general and to HE in particular gravitate around some major aspects. One important and highly recognized contribution of implementing AI is related to the teaching–learning process; under its complex understanding, all aspects that people need to pass through in order to acquire new knowledge and skills and which ultimately influence their attitudes, decisions, and actions [10]. It involves going from "what" to "how", from" how" to "why", and "what for". It is believed that AI can facilitate learning and it can offer both students and teachers a personalized approach, particularly in the case of one-to-one tuition, which has become expensive and out-of-reach in different countries or in contexts where a shortage of qualified teachers has been noted [6,7,11]. The tools offered by AI have proven to be effective, and some have become widely used in universities and high schools in Western countries, such as virtual and augmented reality, voice assistants, etc., just to mention a few of the many other applications [9]. Technology and innovative tools and solutions are emerging with the purpose of identifying the gaps in learning and improving pedagogical methods with the aim of achieving academic success. The educational sector benefits from the AI curriculum, which enables students to progress faster and more efficiently so that their learning goals are achieved at a pace in accordance with the ever-changing demands of the 21st century [7]. AI can provide students with different resources such as translation tools, voice assistants, chatbots, VR and gamification, personalized tutoring and studying programs, instant assessment, and feedback. Such benefits range from developing global classrooms for various

types of students and addressing different learning needs to creating opportunities "to combine ideas across scientific boundaries" ([12], p. 20), being more inclusive, and offering a more personalized education. The COVID-19 pandemic is an example of the role that new technologies can play in an unpredicted situation. The sudden immersion of most of the world's educational systems in the online mode proved to be less difficult and had fewer negative effects in those countries with a higher degree of digitalization on those students and teachers who were already familiarized with new technologies and on those institutions that were already connected and equipped with new technological tools [3,4].

The benefits of the implementation of AI in HE are appreciated in relation to the research process. AI offers exceptional opportunities to increase interdisciplinary, multidisciplinary, and transdisciplinary research, as AI facilitates searching through a huge number of sources, selecting eclectic topics, transferring methods from one field to another, or mixing research methods when searching complex topics [2,13–15]. By collecting and processing big data, facilitating collaborative research, and smoothing the flow of communication between researchers, new avenues for research are opened, new ideas circulate, and new solutions become easier to identify and apply [14].

The efficiency of HE institutions is another area where AI is considered to be able to have a large positive impact. From the enrollment process to the functionality of a HE institution, there are a lot of activities to which the presence of AI could bring more efficiency and better security [6,14]. As HE is thriving for internationalization, in all aspects (students, faculty, and curriculum) ([16], p. 62), it needs to develop dynamic and competitive strategies to make the enrollment process more accessible and more efficient, provide students and staff with better facilities, and provide more security. There are examples of universities that already employ chatbots for marketing to provide personalized help and guidance for students [17], reduce repetitive tasks such as preparing lesson plans or assessing quizzes [6,18], use "enrollment analytics" to reach out to as many students as possible, and allocate financial resources and distribute different facilities ([19], p. 2).

Far behind the benefits brought by implementing AI in HE is the need for AI required by the market. As HE is asked to provide better workers, better leaders, and better citizens, the challenges induced by technological advancements have to be internalized by the educational process in order to respond to the new working environment. New jobs are expected to appear in the next decade, and for that, new skills and competencies have to be built. In a report prepared by Dell Technologies ([20], p. 3), it was stated that schools should teach how to learn instead of what to learn, as the ability to acquire knowledge will be at least as important as the knowledge itself. The education process needs to offer future workers the necessary skills to master technology and to adapt to technology quickly. Therefore, those who are less exposed to new technology, including AI, will struggle more in an increasingly technologized world, will be less productive, and fewer opportunities will emerge for their careers.

### 2.2. Challenges of Implementing AI in HE

However, the claims that AI brings improvements to the classroom per se are still highly debatable. As a scholar mentioned, "it is still unclear for educators how to make pedagogical advantage of it on a broader scale and how it can actually impact meaningfully on teaching and learning" ([21], p. 1). More and more voices are asking for proper control over new technologies in general and over AI in particular, in terms of close monitoring, rules, and legislation to be issued to avoid breaches of ethics, privacy dilemmas, and biases. Who bears responsibility if an algorithm is incorrect?

AI integration in education is a complicated task that requires a functioning and reliable framework, supporting infrastructural modifications and a significant amount of digital equipment. It also involves a training process for all involved in the teaching–learning process in order to make the system operational, and a strategic vision toward AI implementation needs to be developed at the level of institutions and the system [18,20,22,23]. Adapting to AI is provocative, and it involves both human and financial efforts. Teachers

are required to adapt to the new methods; however, some of them lack the necessary training to perform their tasks successfully as the resources to train them are scarce and have not been previously budgeted. Mastering technology involves hours of training and practicing, independent of their teaching timetable, raising issues of time-management, availability, and double-tasking. "...The question is not how to acquire or use them (new technologies), is how to develop and adapt them to the different realities of multivariable environments" ([22], p. 566). It can be rather challenging and expensive for many HE institutions, leading to educational inequity, as, with the continuous upgrading of software, universities will constantly face the need to adapt to the ever-changing technological solutions and require more funds to keep up to date. This process widens the gap between the heavily funded HE institutions and the ones struggling financially. For instance, countries such as the U.S.A., Finland, China, and the U.K. have launched strategic policies of education in order to integrate intelligent technologies into education, providing HE institutions with grants and resources to develop AI learning platforms and train the academic staff to become familiar with AI [18,24]. These policies could enhance academic success and, at the same time, attract international students due to their innovative approach, while other universities lag behind as they lack the financial resources to adapt to the new requirements [18,25].

Additionally, there is a social dimension to humans that is in question under technological developments, together with the framework that is best for developing specific skills. It is believed that, in the future, the most valuable resource in the digital era will be the innovative mind, creative competences, cognitive skills, and emotional intelligence [17]. Humans need social interaction and a sense of belonging to a community, which are primarily associated with the process of acquiring information in an academic environment that is populated by humans. It requires physical interaction and communication, and machines or software cannot create the typical conditions that can nurture emotional and social intelligence. "Learning is a social exercise; interaction and collaboration are at the heart of the learning process" ([21], p. 4). The main aim of creating social interaction in an educational environment is to encourage "effective communication, cognitive presence–exploration, resolution and construction of understanding" ([26], p. 65). It has also been argued that AI is not yet capable of matching the cognitive capabilities of humans. The processing capabilities of machines can easily be outsmarted ([27], p. 43). The notion of human intellect is continuously challenged, and contemporary societies place great emphasis "on sensitivity to others, on the capacity to cooperate with strangers" ([28], p. 226). While information-processing techniques and computer simulations are undoubtedly powerful instruments in the teaching–learning process, they are not capable of perceiving the whole spectrum of emotions that influence human behavior and, ultimately, the motivation to achieve goals. These trends will bring new challenges that may disrupt the labor market. Machines performing tasks previously handled by humans could "exacerbate the gap between returns to capital and returns to labor ([29], p. 93). Apple CEO Tim Cook commented that if he were in the position of a country leader, he would set the goal "of monopolizing the world's talent", leading to a completely different job market. Low skills and low-income jobs will be replaced by machines, whereas jobs generating high income and requiring specific advanced skills and talent will be in high demand. This may also lead to income inequality, which has the potential to disrupt society through increased social challenges, segregation, and marginalization ([29], p. 93).

There is definitely an increasing debate about the ethics behind implementing AI in HE. Responsibility toward the actions of algorithms, chatbots, and robots, the ethics behind those who create AI and those who operate AI, data privacy, and security are big themes that have been launched in the ethics debate about AI [19,23,30]. There are many concerns regarding the fact that students are tempted to cheat as more and more highly accurate software and chatbots can produce works that are required for their academic formation (the passionate debate around ChatGPT, for example), and this temptation, in turn, raises ethical issues. Questions such as how students should be educated about ethics

in connection to AI is an under-researched topic, and more should be achieved in order to further develop ethical behavior. What theoretical approaches about ethics should be embraced when we discuss robot or chatbot responsibilities? These are the questions that have not even theoretically been well defined, and research in this area is already lagging behind due to the speed of technological advancement (not to mention the difficulty of putting these aspects into practice) [30,31].

Drawing from the multi-faceted perspectives presented above, there is clearly a need for a deeper understanding of the impact of AI in HE. Therefore, this research aims to bring more clarity on the benefits and drawbacks of using AI in HE by investigating the opinions of academics from Romania.

## 3. Research Methodology

To investigate the perspectives of academics on implementing AI in HE, qualitative research was employed. As the purpose of this research was rather exploratory, qualitative research was the most suitable methodology to investigate the opinions regarding the research goal [32]. Similar research using in-depth, semi-structured interviews has been conducted on topics related to AI and HE from different perspectives [11,33–35], with the same goal of allowing the participants the possibility "to demonstrate what is important to them" ([33], p. 5).

For the present study, 18 academics from 5 Romanian universities were interviewed between August 2022 and December 2022. A total of 50 academics were invited to participate in the research from different Romanian universities, who specialized in Social Sciences and Humanities, on the basis of convenience. The reason for selecting these areas of expertise for academics was related to the intention to have a wider perspective of academics on the implementation of AI in HE. Out of the total number of academics invited, 18 agreed to respond to the invitation. All aspects regarding privacy and ethics were followed in the process of collecting data. The respondents were over 40 years of age, had solid experience in Higher Education, and had a gender split of 11 women and 7 men. All the respondents were in the field of Social Sciences and Humanities. Two main questions were asked in the discussions: (1) What are, in your opinion, the advantages and the opportunities of implementing AI in Higher Education? (2) What are, in your opinion, the disadvantages and threats of implementing AI in Higher Education?

The opinions expressed by academics were codified by the researchers in two independent groups. Two rounds of codification were performed by each team, and the third one was carried out by the entire team in order to harmonize the codes created. After the codes were established, the research team agreed on themes [36,37]. The interpretation followed the "paper and pencil" approach [38], as the responses were not very numerous and no big data inputs were generated by the responses. The summary of the subthemes and themes that emerged as a result of the codification process are presented in Figure 1.

There are some limits to this research. There was definitely a small sample of academics that were interviewed, and their opinions should not be generalized to all academics from the fields of Social Sciences and Humanities in Romania. The size of the sample is quite limited, and the results should be interpreted from an exploratory perspective as a starting point for further inquiries, qualitative or quantitative, on larger samples. However, as the responses received from these 18 respondents reached saturation, we consider that the sample is sufficient enough to provide useful information on the topic. Additionally, the convenience of the selected sample, despite its limitations, provided accurate information, as all those who accepted to participate in the research were truly interested in and very eager to discuss the topic. Combined with their experience in teaching and research in HE, the results provided by this sample are useful for enriching the literature on the topic. This study does not intend to draw any comparisons based on specialization of academics, gender of academics, or type of universities. Further research can focus on comparative aspects in order to examine in depth what academics consider to be good or not so good about implementing AI in HE. This study is exploratory and adds to the

literature concerning the implementation of AI in HE with an example from a country that faces a sort of paradox: good internet infrastructure but low digitalization.

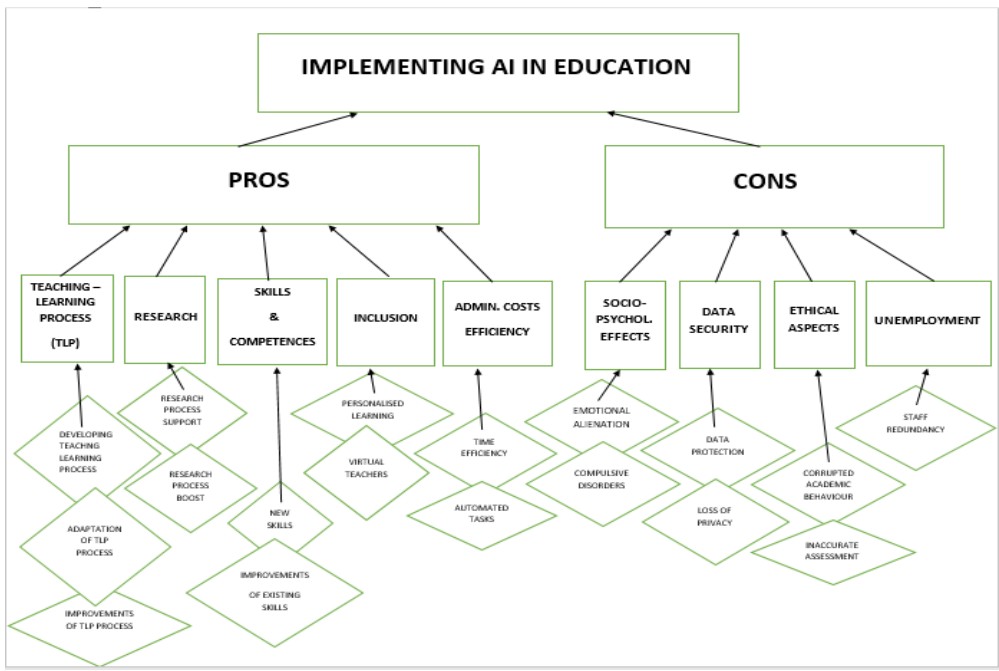

**Figure 1.** Pros and cons of implementing AI in HE.

## 4. Results

### *4.1. Pros of Implementing AI in HE*

There are definitive positive aspects as well as many opportunities identified by the interviewed academics (Figure 1) (except for one academic, who did not acknowledge any benefits of implementing AI in Higher Education). The codes that emerged for the first question's responses have been grouped under the following themes: teaching–learning process (TLP), research, skills, and competences, inclusion, and administrative costs.

***The teaching–learning process*** is by far the most important theme that emerged from the discussions. Sixteen out of the eighteen respondents agreed that, in the future, and with the help of AI, the process of learning will change, resulting not only in a better educational experience and better outcomes but also in new learning content, processes, and methods. Three major directions have been assigned to this theme: *developing the TLP, the adaptation of the TLP*, and *improvement in the TLP*.

*Developing the teaching–learning process* as a positive consequence of the implementation of AI in HE means, in the opinion of the interviewed academics, new content (new curricula development, new content for the existing curricula, and new disciplines), new possibilities of learning (using designated software/apps), and new methods of teaching and assessment, as was suggested in a summarized expression provided by one of the respondents: "fundamental change in what we teach and how we teach" (A1). These are viewed as benefits for students ("they will be better prepared for the new realities" (A3), "they will learn more and they will be attracted to learn different stuff, as foreign languages, because they will have the help they did not have until now. We can imagine that not very far in the future, students will have their own tutor, free of charge, which will be at their disposal at any time, day or night, willing to explain, again and again, the same thing until the student understands it. Nothing will be too difficult, too embarrassing or too useless to learn or to start to learn, when you have somebody that you will trust, that you will feel without sentiments (no risk to patronize you or to feel that they patronize you) (A15)".

*Adaptations of the teaching–learning process* are also mentioned by almost all of the interviewed academics. Under this aspect, the following codes related to personalized

learning have been included: adaptation of the learning process to students' needs, specific tutoring systems for students, flexibility, and curriculum customization. "AI will offer the possibility to adapt to students' needs, in terms of time, and learning resources" (A7).

*Improvements in the teaching–learning process* are the third subtheme. Codes related to better teaching assessments and evaluation methods and better time management for learning and assessing have been associated with this theme. "Using different tools in the teaching process" (A10), "making the learning process more accessible" (A8), "more attractive teaching methods and more meaningful for the students' assessment and evaluation" (A3) are a few examples of the comments provided by the interviewed academics.

The second theme considered is ***research.*** Almost half of the respondents (eight out of eighteen) agreed that the involvement of AI in the research process would be beneficial. Two main subthemes emerged under this theme: *support for the research process* and *boost of the research process*. Codes such as "research assistant" or "help in analysing big amount of data"," structuring information", "categorizing information" "conveying accurate data", and "processing data with the help of software" were mentioned by the respondents. The second subtheme, *the boost of research*, was highlighted, and the academics mentioned opportunities for cooperation between international researchers, enabling faster and wider peer review, the identification of new research avenues, and increases in holistic and interdisciplinary, multidisciplinary, and transdisciplinary research. "Imagine the help you can count on at any time: somebody or better, something, that will look into the literature and will categorize for you different resources: article, books, press release, magazines . . . will organize them as you want, by date, by citation, by . . . I do not know . . . and next week, because you realize that you need something else, it can change that organization, using more filters, new criteria and so on . . . it is not necessary to explain too long what to do, it is not necessary for you to spend hours and days doing this . . . " (A9). "AI has already proved to be useful in translations and this is extremely helpful, not only as an assistant to translate into accurate English, suitable for academic publishing, but also for connections. You may connect, with the help of AI, breaking the language barrier, with a researcher from Japan or from India, without being necessary for you or for them to be proficient in a tertiary language (usually English) or in their language. You may speak in Romanian, exposing your idea very clearly, she/he can speak in Japanese, and AI will do the job of an accurate, instant translation. Time is saved, the idea is saved and new research opportunities are created!" (A6).

Eight respondents out of the total of eighteen highlighted that, if AI (and digitalization in general, as one of the respondents underlined) is correctly implemented and aligned with human-oriented goals, the future workforce will benefit from a wide range of skills adapted to the requirements of the emerging labor market. Consequently, two subthemes emerged under the theme of ***positive effects on skills and competences*** here: one is related to *improvements in the skills and competences* that students gain throughout their academic instruction (mostly digital skills, which will be improved) and *new skills and competences* as a result of technological evolution in general and AI implementation in particular. In the area of improvements, all of the academics who mentioned this aspect looked to the digital skills ("others than those of surfing social media and checking in new locations and parties" (A2). Digital skills have become a must for the digital economy that we are going toward. "Everything in the future will be "intelligent": the house, the car, the school . . . you have to figure out how to use all these intelligent devices, how to work with them" (A18). Related to new skills and competences, academics mentioned that implementing AI in HE will contribute to new business ideas, new technologies, and new products that will contribute to the progress of mankind: "maybe, with the help of AI we will find the cure for cancer, or the solution to pollution . . . I do not know . . . " (A13). "Somebody needs to build the intelligent robots, and this is new . . . to create the artificial brain is new. Watch the Bicentennial Man movie . . . " (A12).

The third theme that emerged was ***inclusion.*** Almost a third of the respondents mentioned easier access to education for the more challenged students. With the help of

AI, students with different needs and backgrounds will have the opportunity to continue their academic development due to specialized methods of learning that are adapted and tailored to suit their needs. "There is a big problem today for those who have to walk kilometres to school, because the access is difficult, the professors are missing or because the students have difficulties in reaching out an education facility. With the help of AI, remote education should not be a problem and many people can continue their academic voyage in a less formalized pattern" (A11).

The fourth theme identified is related to ***administrative cost efficiency***. "Time efficiency" code was mentioned by six respondents, in comparison with the "cost efficiency/reduction" code, which was only mentioned by two respondents. In this case, the enrollment process is regarded as the most suitable for cost reduction. Over a third have agreed that administrative tasks can be automated, and this can result in improving the flow of information between students and administrative staff, "speeding the registration process" and the "processing of great amounts of data", and "reducing bureaucracy" for the benefit of students.

There is one respondent who considered that there were neither benefits nor opportunities in implementing AI in HE, at least not in the near future. In the opinion of this professor, AI is not sufficiently regulated, and the ethical aspects are too complex and less investigated. The consequences of using AI in the education process can be seen after a significant period of time and are not yet properly investigated in relation to critical thinking, building and developing moral values, or sharing responsibility. "If a human makes mistakes, the human is responsible. The actions you make, as a person, are the actions you bear the responsibility for, and you are aware of the consequences of your actions. If AI makes mistakes, who is responsible?" (A14).

### 4.2. Cons of Implementing AI in HE

The second research question looked into the opinions regarding the negative aspects of implementing AI in HE. The weaknesses and threats that could be inferred from the respondents' answers fall into the following themes: socio-psychological, security, ethical, and unemployment.

The main concern that respondents expressed was the fact that, due to the extended time spent in the virtual environment, students (and not only them) could be ***socio-psychologically*** affected. Two respondents mentioned "addiction", and ten respondents agreed on the "critical level of social interaction" due to "isolation", which can lead to "alienation", a "lack of empathy", "decreased emotional intelligence", and a slowed process of exchanging and comparing ideas. "Lack of motivation" and "barriers in communication" have emerged as the negative consequences of human–machine interactions. In the opinion of six respondents, replacing traditional teaching methods with the so-called innovative ones can result in demotivated students and professors, who can experience professional decline. Here are some examples of the comments provided for this theme: "The lack of human interaction, with the negative psychological effects that specialists consider, superficiality, fake results in research, the tendency to consider the human (the teacher) almost useless" (A3). "The danger of becoming addicted to a certain product, no longer using your own critical filter" (A1). "In the absence of a teacher, in an assessment test, for example, AI cannot perceive what a teacher can observe: for example, the health of a student. If a student gives a wrong answer, it is recorded as such, but if the teacher notices that the student is not feeling well and still participates in the test, the teacher may consider postponing the test in the case of the sick student. So, since the human component is missing, the results of such assessment can be misinterpreted" (A4).

***Security*** is the second theme that emerged from the discussion with the academics. "Data protection" was mentioned as one of the most sensible aspects of working with AI in HE institutions. "Data storage security", "confidentiality", and "loss of privacy" were regarded as the most exposed elements to hackers and those who have criminal intentions in the virtual environment. Ten respondents were concerned about the above-

mentioned aspects of digitalized learning, with three of them underlining the possible disregard of fundamental human rights, thus breaching the ethics of the educational system. The findings reveal the fact that AI is believed to be easily hackable and that data can be leaked or stolen to be used later for identity theft, bullying, or discriminating. Therefore, the participants in the study advocated for AI safety research and modernized and adapted laws.

*Ethical aspects* are also a concern. The codes "cheating", "responsibility", and "inaccurate assessment" are the codes that led to this. Nine participants agreed on the possibility of cheating during exams or tests, as AI provides students with so many temptations (from writing the assignments to using first-rate devices to inform oneself during the exams or tests). Superficiality during the process of accumulating information might lead to attempts to cheat, and this can easily convey inaccurate assessment results. In the long run, this will mainly affect the graduates because they lack the skills they should have gained through their academic program, but also the universities issuing the diplomas, which should be the solid bona fides that employers need to select the best candidates for their vacancies. One of the professors interviewed mentioned this aspect quite clearly: "As long as AI is used for ethical purposes, I see no disadvantages or threats" (A3). "More and more sophisticated software tempts and can instigate to fraud. Look at ChatGPT . . . " (A10) "Implementing AI in HE cand lead to an increase in students' unethical behaviour if AI encourages replacing student-written texts with algorithm-written texts" (A11).

The last theme grouped codes under the topic of *unemployment*. Five of the interviewed academics expressed concerns about the prospects of "being replaced" by silicon-based knowledge providers. In addition, the staff involved in administrative tasks are highly likely to be made redundant ("staff reduction/replacement") in the near future, according to the opinions expressed by the academics.

Two of the respondents did not find any drawbacks to implementing AI, and two professors did not consider the process of AI implementation threatening.

## 5. Discussion

The responses of the academics are in line with the results expressed in the literature. On the positive side of AI implementation in HE, as expected, the *teaching–learning process* is considered to be the most influential aspect of HE. Many gains are encountered in terms of improvements, adaptability, customization, flexibility, and so on. These are also the most common positive aspects revealed in the literature [6,7,11]. As in the studies cited by this work [10], the interviewed Romanian academics see the implementation of AI in HE as a solution for some of the problems that the education process encounters today and as a way to better respond to the needs of a generation that is born and is already connected to new technologies. The *research* area is also present when positive aspects related to AI are mentioned. As anticipated, AI can serve as an excellent research assistant that can save time and effort, or an excellent translator (free of charge), permitting the dissemination of research results to a worldwide audience: a platform capable of connecting and facilitating international collaboration, interdisciplinary work, and so on. The gains in *skills and in* are also widely discussed, not only in the academic literature [10], but also with the media and by civil society and policymakers, and it was an expected result to be identified among the positive aspects of implementing AI in HE. It is not new for anybody that the future jobs will ask for different skills and competences [20] (p. 3). Similarly, the topics of *inclusion or cost efficiency* have been emphasized as advantages of implementing AI in HE.

Some benefits were not explicitly mentioned, such as lifelong learning, and others were expressed as possibilities that were largely open to everybody. Personal experience with AI was very rarely expressed in terms of "I have experienced this benefit of AI" (except for a very few related to translation and the foreign language learning process). This can be associated with the fields that the sample of academics belongs to, namely Social Sciences and Humanities, as well as with the fact that Romanian universities started to realize the importance of being up-to-date with new technologies during the pandemic. The sudden

immersion of the educational process in the online mode in March 2020 demonstrated that neither students, universities, nor teachers were prepared to operate online teaching platforms (not to mention other types of new technologies) [3,4,39,40].

On the negative aspects regarding the implementation of AI in HE, the opinions of the interviewed academics are also in line with the literature findings concerning the psychological effects (social interaction, data protection, and privacy) and ethical aspects.

However, some aspects are missing from the opinions expressed by the academics but are highly debated in the literature. One of these aspects is related to the effort that both academics and HE institutions need to put in to keep updated with AI, meaning training, financial, psychological, and psychical resources that need to be employed [18,20,22,23]. This lack of concern can be explained by the absence of a serious discussion about what, in fact, implementing AI in HE means. It also expresses little knowledge about what AI is, and a lack of understanding of what the implementation process of AI in HE entails. The absence of this discussion is in line with a lower level of digitalization of the Romanian economy and a lower penetration of AI in the process of education in Romania [3,4]. This conclusion is of great importance because it can lead to a sort of complacency (academics consider themselves either well prepared for AI, or AI is less difficult to work with and too costly for them to prepare for AI), with negative effects on HE competitiveness. It is absolutely necessary for HE managers to consider that being contemplative or indifferent toward AI (from the academics' side) is similar to opposing AI per se, and this can lead their institution to a competitive disadvantage. Students who were born in the digital era will not accept a return to the traditional model of education just because academics tell them to do so or just because academics are not well prepared for the new technologies. Students will look for those universities that are well connected to technologies and are capable of training them to master these technologies. Implementing new technologies in education is about "how people use technology to express their identities and connect with each other around that identity" [41].

Another aspect revealed by this research is the ethical aspect of AI. The major concerns expressed by the interviewed academics are related to cheating and the assessment process, which may be flawed due to a lack of human intervention. This is a very narrow perspective on the ethics of AI. Only one opinion (and this was quite vaguely expressed) related to the difficulty of adopting suitable theoretical approaches toward ethics that should be taught regarding AI. No opinion is linked to the need to have an AI implementation strategy in HE, where ethical aspects as well as the objectives that HE institutions target through the use of AI are explained. This means that AI in HE institutions in Romania, on a large scale, is still in its infancy. Further, this suggests the view that there is no strategy regarding implementing AI debated among academics, and this is, again, in line with the lower level of digitalization of the Romanian economy.

## 6. Conclusions and Recommendations

This study highlights the perspectives of academics regarding the implementation of AI in HE. The study is more exploratory in its sense, and the qualitative dimension of this research should be interpreted rather as a starting point for a comprehensive study employing a representative sample. However, the opinions expressed by these eighteen academics revealed that the implementation of Artificial Intelligence in Higher Education in Romania, is more likely to be seen as a future process than a reality that academics are confronted with. The opinions expressed are more general, conveying a rather minimal exposure to AI. This is an expected result, as Romania is a paradox: it is a country with one of the highest internet speeds in the world but with a low-level digitalized economy (education included).

The present study adds to the literature that investigates AI implementation in HE mostly as empirical research, with a case coming from a country (Romania) with a less digitalized economy but with competitive access to the internet. Our study supports the results obtained by other researchers in terms of the positive and negative consequences of

implementing AI in HE. Most of the perceptions regarding the pros are related to the effects on the teaching–learning process, research, and the development of new skills. As for the negative aspects, socio-psychological effects and the loss of the sense of "being human", a kind of fear of the dissolution of humans as social beings, together with security and ethical aspects are among the most prominent concerns coming from the sample of academics investigated in this study. The opinions, however, do not expose concerns regarding the efforts (human, time, and financial) that are necessary to keep up with the development of new technologies. The lack of concern about these efforts can be explained either by their little knowledge about what AI means (from an implementation point of view) or by a lack of understanding of the place that AI holds and will have in the near future in education. Nonetheless, there is a lack of concern regarding the fact that these efforts to adapt constantly to new developments can lead to inequalities among HE institutions, between those with resources and those lacking in resources. Just thinking of how to fight against students' temptation to cheat because of ChatGPT, for example, or how to restrict the implementation of AI, reflects a narrow perspective toward AI and a lack of understanding of both the real potential and the real danger that chatbots or algorithms may pose. The ethical aspects are far too concentrated on cheating, and they would rather be concerned with the efforts and strategies that should be dedicated to eliciting an ethical approach to AI. The need to develop a strategy regarding implementing AI in HE is missing from the opinions expressed by the interviewed academics. The disruptive effects that new technologies may create on the labor market, as some repetitive jobs will disappear, are not among the important concerns of those interviewed. This can be interpreted as a concern about one's own employability and not as a preoccupation with the employability of others in other industries.

## 7. Research Limitations and Future Research Directions

One limitation of this study comes from the fact that the results cannot be generalized to the whole population, considering the study's qualitative nature. However, the conclusions of this study are helpful from practical perspectives, as they may be beneficial to decision makers in HE institutions when policies regarding human resource training and recruiting are considered, when curricula development is discussed, and when strategies for the development of Higher Education institutions are designed. AI is here, and we have to deal with it. Educators need to be involved in implementing AI; they need to be trained, and they need to be aware that this is a continuous process. These avenues can represent suitable future research directions, especially the appraisal of the connections between them. Another limitation of this study can be derived from the number of employed angles. Other variables should probably be included in the model to offer a more thorough view of the impact of AI on HE. Nonetheless, the findings of this study show that managers need to be aware that a strategy is necessary to manage all these efforts, to grasp the benefits, and to educate both students and academics about this new reality. Strong ethical policies need to be developed at the level of institutions and society, and a wise management of human resources in HE is required.

**Author Contributions:** Conceptualization, R.M.Z., A.I.P. and R.Z.; methodology, R.M.Z. and T.E.; analysis, A.I.P., T.E., R.M.Z. and R.Z.; writing—original draft preparation, A.I.P., T.E., R.M.Z. and R.Z.; writing—review and editing, A.I.P. and R.M.Z. All authors have read and agreed to the published version of the manuscript.

**Funding:** This research received no external funding.

**Institutional Review Board Statement:** Not applicable.

**Informed Consent Statement:** All the participants consented to participate in the research and a consent form was elaborated specific for this research.

**Data Availability Statement:** We used data collected from the participants.

**Conflicts of Interest:** The authors declare no conflict of interest.

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
