# Peer review of "Implementing Artificial Intelligence in Higher Education: Pros and Cons from the Perspectives of Academics"

_societies, doi:10.3390/soc13050118_

Round 1

Reviewer 1 Report

1. What is the main question addressed by the research?

The article addressed the issue of implementing AI in HE, an actual and very debated issue in the literature, but less investigated in the Romanian context.

2. Do you consider the topic original…

Yes, the topic is original, mostly because it approach  it in a context that is less investigated. The exploratory character of the research is interesting from my point of view.

3. What does it add….

From my point of view, the idea of the study is quite original. The interpretation of the results confirms other studies’ conclusions and adds as empirical evidence to the literature. It confirms, from my perspective, the need to use qualitative methods for investigating this kind of topic.  

4. what specific improvements….

The author(s) can explain more how the size and the selection of the sample affect the final results.

5. Conclusion are consistent…

Yes, the conclusions are consistent with the evidence and arguments presented in the paper.

6. References…

Yes, the references are sufficient. I recommend, however, the study of Aivaz, K. A., & Teodorescu, D. (2022a). College Students’ Distractions from Learning Caused by Multitasking in Online vs. Face-to-Face Classes: A Case Study at a Public University in Romania. International Journal of Environmental Research and Public Health, 19(18), 11188 that may be helpful for the research.

Final comments for the author(s).

The article addressed the issue of implementing AI in HE, an actual and a very debated issue in the literature, but less investigated in the Romanian context. The topic is original, mostly because it approaches it in a context that is less investigated. The exploratory character of the research is interesting from my point of view. The idea of the study is quite original and the interpretation of the results confirms other studies’ conclusions and adds as empirical evidence to the literature. It confirms, from my perspective, the need to use qualitative methods for investigating this kind of topic. The conclusions are consistent with the evidence and arguments presented in the paper. However, the author(s) can explain more how the size and the selection of the sample affect the final results. The references are sufficient. I recommend, however, the study of Aivaz, K. A., & Teodorescu, D. (2022a). College Students’ Distractions from Learning Caused by Multitasking in Online vs. Face-to-Face Classes: A Case Study at a Public University in Romania. International Journal of Environmental Research and Public Health, 19(18), 11188 that may be helpful for the research.

It is an interesting topic and the research is well designed and nicely written. It can be useful, for strengthen the argument, to refer to some articles that  debate related aspects as:

Aivaz, K.A., & Teodorescu, D. (2022). College Student's Distractions from Learning Caused by Multitasking in Online vs. Face-to-Face Classes: A Case Study at a Public University in Romania. International Journal of Environmental Research and Public Health, 19(18), 11188.

Aivaz Kamer-Ainur, Teodorescu Daniel, The Impact of the Coronavirus Pandemic on Medical Education: A Case Study at a Public University in Romania, in  SUSTENABILITY, Volume 14 Issue 1. Published:  2022 eISSN:2071-1050. DOI: 10.3390/su14010542

https://www.mdpi.com/2071-1050/14/1/542

Author Response

Dear Reviewer,

thank you for your input related to our paper. We revised the part dedicated to aspects related to the influence of the size and of the selection method on the final results, as it show the revised version of the paper. 

Also, we have introduced in the references the articles suggested. 

Reviewer 2 Report

Abstract:

Should clearly and succinctly articulate findings. 

Should state epistemological stance the paper adopts 

Should state study contributions (theoretical and practical)

Introduction:

Sentence construction is a major challenge consequently, the argument is lost in language translation. Consider using a professional English language editor. See sentences 80-95; 131-180, these are well articulated.

Theoretical background:

This section is congested, the use of sub-headings can aid readability of the document 

AI is the main topic and HE the context, the literature review should extensively cover both and merge them seamlessly. 

Discussions on labor market and school causes confusion on the actual context of the study - study must be focused on its context HE

Methodology:

Qualitative sample are determined by saturation and not number of participants who accepted the invitation. 

Results"

This section is text-heavy and use of figures/diagrams will aid the pictorial presentation of the document

Conclusion:

Theoretical implication are omitted. 

The practical implication are superficial and need to be reinforced.

Author Response

Dear Reviewer,

thank you very much for your time for reviewing our paper and for the suggestions provided. We hope that our article looks better now. 

We have improved the abstract according to your suggestion and state, explicitly, the findings of our research, the epistemological stance and the contribution  to the literature.

We have checked the English with the help of an official translator.  

We add sub-headings. 

We explain more about the context.

We contextualized the mention about labor market in relationship to HE

We completed the methodology.

We have highlighted in the part of the results the opinions in order to structure better the idea. 

We underlined in the conclusions the empirical and practical contribution of the study. 

Round 2

Reviewer 1 Report

I am happy  with the changes.

Reviewer 2 Report

I recommend use of professional English language editors for future publications. They add a lot of value to ones articles, a language certificate must also be provided